# LC–MS Based Lipidomics Depict Phosphatidylethanolamine as Biomarkers of TNBC MDA-MB-231 over nTNBC MCF-7 Cells

**DOI:** 10.3390/ijms232012074

**Published:** 2022-10-11

**Authors:** Alan Rubén Estrada-Pérez, Norbert Bakalara, Juan Benjamín García-Vázquez, Martha Cecilia Rosales-Hernández, Cynthia Fernández-Pomares, José Correa-Basurto

**Affiliations:** 1Laboratorio de Diseño y Desarrollo de Nuevos Fármacos e Innovación Biotecnológica, Escuela Superior de Medicina, Instituto Politécnico Nacional, Plan de San Luis y Díaz Mirón, Ciudad de México 11340, Mexico; 2Centre National de la Recherche Scientifique, École Nationale Supérieure de Technologie des Biomolécules de Bordeaux INP, Univeristé de Bordeaux, 146 Rue Léo Saignat, 33000 Bordeaux, France; 3Laboratorio de Biofísica y Biocatálisis, Sección de Estudios de Posgrado e Investigación, Escuela Superior de Medicina, Instituto Politécnico Nacional, Ciudad de México 11340, Mexico

**Keywords:** breast cancer, lipidomics, triple negative breast cancer, LC–MS

## Abstract

Breast cancer (BC) is the first malignant neoplasm in women, with a high death rate despite early diagnoses and treatment advances. Significant differences exist between the most common BC and triple-negative breast cancer (TNBC). TNBC presents molecular differences such as lacking expression of the estrogen receptor (ER), progesterone receptor (PR), and HER2 proteins, making this cancer have a poor clinical prognostic and lack clear strategies for its treatment. However, growing evidence points to metabolic dysregulation as another differential process between stages and types of BC. Therefore, the study of this crucial hallmark could identify new therapeutic targets to treat this aggressive form of BC. These differences induce an in vitro exploration of the metabolic behavior of the MCF7 cells (nTNBC) and MDA-MB-231 (TNBC) cells under lipidomic based LC–MS. The results show more significant differences in lipid regulation (phosphatidylethanolamine) that could be associated with the aggressiveness and difficulties of the treatment of TNBC.

## 1. Introduction

Breast cancer (BC) is the most diagnosed malignancy worldwide. In 2020, the International Agency for Research on Cancer (IARC) estimated an incidence of about 2.3 million female patients [1]. Despite scientific efforts, BC is the first cause of cancer-related death in women in Mexico and the world, causing 684,996 deaths in 2020 [1], probably due to the lack of clear strategies for its treatment, besides its high cost and the several side effects affecting the patient’s quality life [2].

BC is a multifactorial disease involving genetic (DNA modifications) and epigenetic alterations that allow cancer cells to replicate and survive under uncontrolled growth [3] through the activation of several intracellular pathways [4]. There are different types of BC; the most common are grouped here as non-triple negative breast cancer (nTNBC) and triple-negative breast cancer (TNBC). The TNBC lacks expression of the estrogen receptor (ER), progesterone receptor (PR), and human epidermal growth factor receptor 2 (HER2) protein, while nTNBC only lacks HER2 [5,6]. There are no clear pharmacological options to treat TNBC, so its prognosis remains poor [7,8]. Nevertheless, there are other molecular mechanisms involved that could explain the treatment difficulties and poor prognostics, such as metabolic alterations. The insufficient biological data related to metabolic dysregulation in BC make it challenging to design and guide the drug treatment of TNBC. Some of these small molecules, which are very important for carcinogenesis, cancer growth, and migration, including metastases, are lipids [9,10]. They not only act as an energy resource and membrane components but also regulate oxidative stress in the cell [11].

Novel characterization chemical techniques, such as liquid chromatography–mass spectrometry (LC–MS), allow studying proteins (proteomics) and metabolites (metabolomics), including lipids present in cells [12]. LC–MS are studies capable of showing the lipid levels, named lipidomics, of cell cultures [13,14,15]. Differences in sample preparation, extraction methods, LC–MS platforms, and data processing make it a challenging task for a single study to cover the entire lipidomic profile, so new contributions might help to fill the gaps [16,17,18,19]. Furthermore, due to TNBC’s molecular complexity, in this work, we have used LC–MS to identify the dysregulated lipids in nTNBC and TNBC. In this study, two of the most studied cell lines were selected: MCF-7 was chosen as a nTNBC model because it is classified as a luminal A subtype of BC, a subtype associated with a good prognosis and low aggressiveness in contrast to TNBC (represented by MDA-MB-231), which is located in the opposite side of the prognosis spectrum within BC subtypes for its bad prognosis due to its unresponsiveness to hormone treatment and aggressiveness [8,20,21]. Through this work, the authors intend to contribute to the identification of possible biomarkers and differential therapeutic targets, related to the aggressive form of BC, that could guide the treatment of this disease.

## 2. Results and Discussion

In order to find lipids possibly involved in BC aggressiveness, a comparison between two BC cell lines, with diametral differences in both prognosis and aggressiveness, was made; MCF-7 was considered as the model for positive prognosis, low aggressiveness, and low invasiveness and MDA-MB-231 for bad prognosis, high aggressiveness, and invasiveness [20,21,22]. Differences in lipid dysregulation of MDA-MB-231 compared to MCF-7 might be related with the aggressiveness of the corresponding BC subtype.

Although it has been reported that TNBC shows a low percentage (10–15%) of incidence in BC, It draws researcher’s attention due to its poor medical prognosis as a consequence of its lack of response to hormone treatment, limiting the therapeutic options available for its treatment [23]. Due to TNBC’s biological complexity, it is pivotal to gather all possible information to better understand this subtype of the disease, suggest new targets for treatment, and to guarantee the success of BC eradication, while avoiding chemotherapy resistance in the process [24]. However, to suggest novel molecular targets, it is required to find molecular differences at the metabolic level; particularly, lipids are an attractive target because cancer cells increase their use as an energy source, building blocks required for cell division, and signaling molecules, among others [25].

Lipidomic studies could depict potential biological targets [26] for drug design [27] to avoid cancer cell growth, metastases, and chemotherapy resistance. This work is focused on exploring lipidomic differences by using an LC–MS approach to compare nTNBC (MCF-7) and TNBC (MDA-MB-231), as has been achieved for glioblastoma [28]. Our main objective was to identify lipids in TNBC that are possibly related to its poor clinical prognosis and its difficulties in being treated by comparing its lipidomic profile against nTNBC. 

First, as part of the assessment of quality of our data, we observed the formation of three distinctive and discrete groups belonging to QC, MCF-7, and MDA-MB-231 samples, which demonstrated that both cell lines showed marked differences among their lipidomic profiles (Figure 1). By looking at QC samples, we confirmed the LC–MS system’s stability during data acquisition and ensured the validity of the analysis.

The number of different entities present in each cell line are shown in a Venn diagram (Figure 2). We found 505 entities for MDA-MB-231 and 534 for MCF-7, from which 436 were shared among both cell lines. Differences may be due to variations in entities’ concentrations instead of unique presence in a cell line; variations in metabolite concentrations give each cell line its characteristic phenotype.

Figure 3 summarizes the statistical differences in features’ abundances by comparing MDA-MB-231 versus MCF-7 cells and, at the same time, allows us to quickly identify those features that differ the most with at least four-fold changes, as well as the direction of their dysregulation. We found dysregulation of lipids from eight different types, half of them from the glycerophospholipids class (Table 1). Phosphatidylethanolamines (PE) were the most dysregulated lipid type with a mixed tendency (seven upregulated and nine downregulated).

As multiple studies have demonstrated, dysregulation of metabolism is considered one of the hallmarks of cancer in general [29], and aberrant metabolism of different lipids classes is among the main metabolic disorders in cancer cells [30]. The importance of phospholipid in metabolic alterations within BC has been explored in a few works [31,32,33,34], and it seems to correlate with our findings. Phospholipids synthesis is frequently increased in BC, and it has been related to oncogenesis and tumor progression [31]; particularly, an increase in PE plays a major role in some BC tumors [31,32]. BC cells even show increased PE levels (along with diacylglycerol or DAG) as an adaptability response to stress conditions (serum deficient media), showing the importance of this type of molecule in cancer survival [32]. Contrarily, Kim et al. (2016) have reported downregulation of PE [35] by comparing MCF-7 and MDA-MB-231 against MCF-10A (considered a healthy breast cell line). Moreover, an increase in the expression of enzymes and proteins such as Pcyt2, PEBP4, and Etnk-1, involved in phospholipid metabolism, has been reported, proving that these proteins involved in PE metabolism might be useful as novel targets for cancer treatment [32,33,34]. Our mixed results in the direction of dysregulation showed that although a few lipids belonging to this compound class are indeed increased in MDA-MB-231, some others are downregulated, and this might be related to the differences in their sensitivity and response to treatments. Their specific role remains to be further explored.

Along with PE, sphingolipids and, particularly, ceramides, were among the most dysregulated lipids found in our study. In general, sphingolipids are considered as some of the most relevant lipids, since they are used as messengers for cell signaling. Purwaha et al. (2018) observed that accumulation of ceramides in tumors function as a signal for the promotion of TNBC progression [36], pointing at the importance of the study of ceramides and its relation to TNBC aggressiveness. In the sphingolipids class, we only found a downregulated sphingomyelin (SM), which corresponds with the findings from the aforementioned work, in which the authors correlated the presence of TNBC tumors with low levels of SM with a negative disease outcome, so it seems that SM plays an important role in BC aggressiveness [36]. Many more efforts have been made to characterize variations, not only by comparing TNBC against other BC types, but within TNBC, such as the work from Xiao et al. (2022), who combined metabolomic, lipidomic, transcriptomic, and genomic data and found heterogeneity within this BC subtype, thus reporting three metabolic subgroups, of which one is characterized by its elevated concentration of ceramides and fatty acids [37], again emphasizing lipid relevance in BC development. Furthermore, some other studies reported a high abundance of triglycerides (TG) in MCF-7 cells comparing TNBC cells (MDA-MB-231 and MDA-MB-436) and sphingomyelins [38]. Recently, for its relevance, SMs have been proposed as prognostic biomarkers of survival in TNBC patients [36].

We were only able to detect a couple of TG, each with a different dysregulation tendency; it is known that TG are the base for fatty oxidation, and it has been found that TNBC cells present an upregulation of this metabolic process to support cancer proliferation, invasion, and metastasis [39,40].

These results made it possible to identify the great differences in lipids, comparing MCF-7 versus MDA-MB-231 cells. Future efforts should be directed towards the identification of metabolic intermediates to determine the metabolic pathways altered among BC cell types; this will allow the definition of new drug targets in search of effective therapies for TNBC. 

As has been stated by other authors, genomics and transcriptomics have the potential to predict a few aspects of cancer, but alone, they are not enough to deal with all the complexity intrinsic to cancer [21]. Although differences in the cancer model selected for the study strongly influence its outcome and conclusions, cultures created from a single cell offer a good but raw and simplified model compared to tumors obtained from patients which are very hard to obtain and categorize; nevertheless, for both alternatives, it has worth the effort to build up our knowledge of cancer as a disease [20].

In conclusion, this work depicts lipids (phosphatidylethanolamine) as potential biomarkers related to TNBC aggressiveness. The specific role of PE needs to be further explored.

## 3. Material and Methods

### 3.1. Cell Culture

MCF-7 and MDA-MB-231 cells were kindly donated by Dr. Gisela Ceballos Cancino (INMEGEN, Mexico City, Mexico). Cell culture plastic ware was purchased from TPP (Trasadingen, Switzerland). Fetal bovine serum (FBS) and Trypsin-EDTA were acquired from Biowest (Misuri, USA). MCF-7 and MDA-MB-231 cells were grown in Dulbecco’s Modified Eagle’s Medium/High Modified (DMEM) with phenol red. Cells were thawed and maintained in the corresponding media and supplemented with 10% FBS at 37 °C and 5% of CO_2_ in a humified atmosphere. The cells were detached using Trypsin-EDTA upon reaching 75% of confluence. Five replicates with 7 × 10^6^ cells were seeded on flasks of 150 cm^2^ and maintained in the same conditions for 48 h. Then, the lipids from cells were extracted as follows.

### 3.2. Lipid Extraction

Cell sonication was performed with a Vibra-Cell VC 130 Ultrasonic Processor (Sonics & Materials, Connecticut, USA); cells scrappers, methanol, and chloroform were purchased from TPP, Honeywell Burdick & Jackson (New Jersey, USA), and Sigma-Aldrich (Toluca City, Mexico), respectively. Ultrapure water was obtained from a Direct-Q 3 system (Millipore, Massachusetts, USA). After incubation for 48 h, culture flasks were kept in an ice bath, cell culture media were discarded, and cells were washed three times with NaCl 0.9 %. Afterward, 1 mL of methanol at −80 °C was added, and cells were scrapped and collected in 2 mL plastic vials in a bath with acetone in dry ice. Scrapped cells were sonicated by applying pulses with a frequency of 40 kHz with an on and off cycle of 5 and 1 s, respectively; this process was repeated four more times. Afterwards, liquid–liquid extraction was made with 250 µL of chloroform, 350 µL of water, and 250 µL of chloroform once more; separation was accomplished by centrifugation at 5000 rpm, 4 °C for 30 min. For lipid analysis, the organic phase was obtained, dried at 30 °C, and stored at −80 °C. For LC–MS analysis, dried samples were dissolved in 150 µL of IPA/CHCl_3_ 80:20 (*v*/*v*).

### 3.3. LC-ESI-MS Data Acquisition

The separation of metabolites was performed in a UHPLC 1290 Infinity II (Agilent Technologies, California, USA) composed of a 1290 flexible pump (model G7104A) and a 1290 vial sampler with a thermostatic column compartment (model G7129B) within a ZORBAX 300-SB 2.1 × 50 mm, 1.8 µm (Agilent Technologies, California, USA). The elution was carried out using ammonium formate 10 mM and formic acid 0.1 % in acetonitrile/water 60:40 (*v*/*v*) (solvent A) and ammonium formate 10 mM and formic acid 0.1 % in isopropanol/acetonitrile 90:10 (*v*/*v*) (solvent B) with the following nonlinear gradient: 32 %B at 0 min, 40 %B from 1 to 1.5 min, 45 %B at 4 min, 50 %B at 5 min, 60 %B at 8 min, 70 %B at 11 min and 80 %B from 14 to 18 min with a 3 min re-equilibration time, and 60.0 ± 0.5 °C. Further, 10 µL of diluent (blank), cell extract, and quality control samples (pool from all samples) were injected with 0.3 mL/min flow rate. The LC–MS system was first equilibrated by injecting 0 µL of blank sample until no chromatographic variation was observed (blank injection), then, 10 µL of blank were injected once and afterwards to equilibrate the LC–MS system to our samples. Quality control samples (QC) were injected until no chromatographic variation was observed. Quality control (QC) samples were injected before and after every five sample injections; the sequence order was randomly assigned for cell extracts (injected in triplicate, pseudo replicates). Mass spectra were acquired using a Q-TOF (model 6545A) with a Dual AJS ESI (model G1959A) through Agilent MassHunter Workstation Software LC/MS Data Acquisition for 6200 series TOF and 6500 series Q-TOF (version B.08.00, build 8.00.8058.3 SP1, Agilent Technologies, California, USA) using the following conditions: positive polarity with capillary voltage 3500 V, fragmentor: 150 V, skimmer: 65 V, gas temperature: 300 °C, gas flow: 11 L/min, nebulizer: 35 psig, sheath gas temperature: 300 °C and sheath gas flow: 12 L/min, nozzle: 0 V and octupole RF: 750 V. A Q-TOF instrument was operated in a 2 GHz extended dynamic range mode at an acquisition rate of 3 spectra/s for signals in the 50–1700 *m*/*z* range.

### 3.4. LC–MS Data Processing

LC–MS raw data were inspected using an Agilent MassHunter Workstation Software Qualitative Analysis (version B.07.00, build 7.7.7024.29 SP2, Agilent Technologies, California, USA) to determine retention time drift tolerance, ionic species, and compound thresholds through QC samples. Batch alignment and extraction were performed with MassHunter Profinder (version B.08.00 SP3, Agilent Technologies, California, USA) through Batch Recursive Feature Extraction (MFE) for small molecules/peptides algorithm using common organic molecules (no halogens) as an isotopic model, allowing a maximum of two charges and only compounds with two or more ions to avoid false positives. To further reduce false positives, only compounds present in 75% of files in at least one sample group (pseudo replicates) and with an MFE score ≥70.0 were retained for further analysis and exported as a profinder archive (.pfa). Chemometric (lipids) comparison between MCF-7 and MDA-MB-231 was performed in Mass Profiler Professional (version 14.9.1, Agilent Technologies, California, USA). First, principal component analysis (PCA) was performed on all samples (entities or features) to be grouped. Afterward, interpretations for compounds present in blank, MCF-7, and MDA-MB-231 were created and filtered by frequency to only retain compounds present in 75% of all samples for each cell line. Features found in both blank and samples were eliminated from the analysis. A moderated t-test, using Benjamini–Hochberg false discovery rate as multiple testing correction, was performed to find statistical differences regarding compounds by comparing MCF-7 and MDA-MB-231 through a volcano plot (*p*-value ≤ 0.01 and fold change higher than 2 or lower than −2). The lipid class of relevant features was assigned using the IDBrowser tool for compound identification against a compound database from Human Metabolome Database (https://hmdb.ca/, accessed on 15 February 2022), and only those compounds identified with a database search score higher than 60.0 were retained.

## Figures and Tables

**Figure 1 ijms-23-12074-f001:**
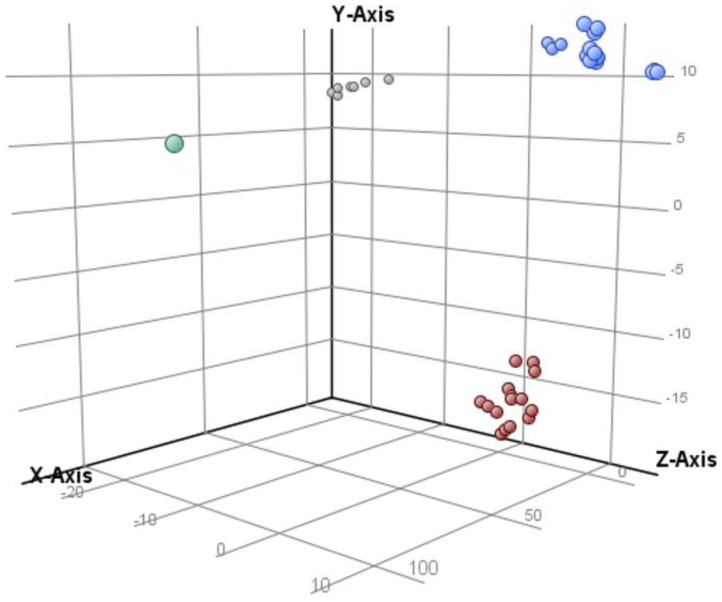
Principal component analysis on all entities and all samples. Component 1: 40.03% (x-axis), component 2: 17.34% (y-axis), and component 3: 11.88% (z-axis). Blank (green), MCF-7 (blue), MDA-MB-231 (red), and QC (gray).

**Figure 2 ijms-23-12074-f002:**
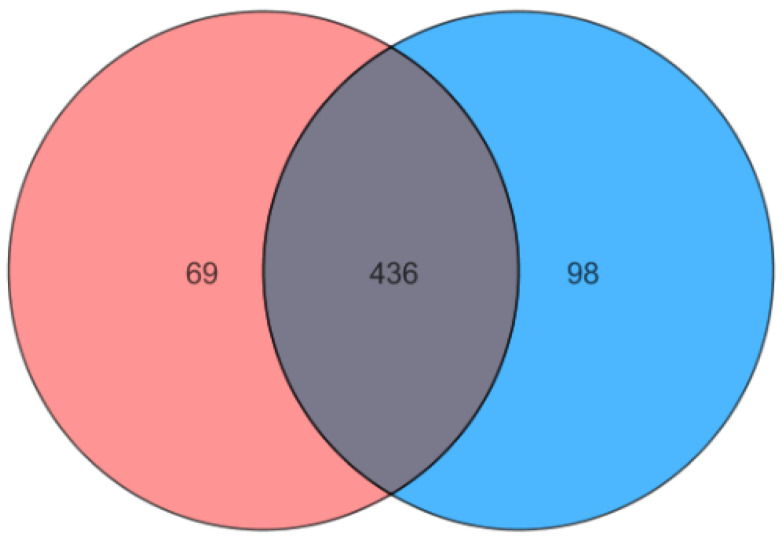
Venn diagram of MDA-MB-231 (red) and MCF-7 (blue) compounds. Both cell lines share 436 compounds (gray); 69 and 98 are found only in MDA-MB-231 and MCF-7, respectively.

**Figure 3 ijms-23-12074-f003:**
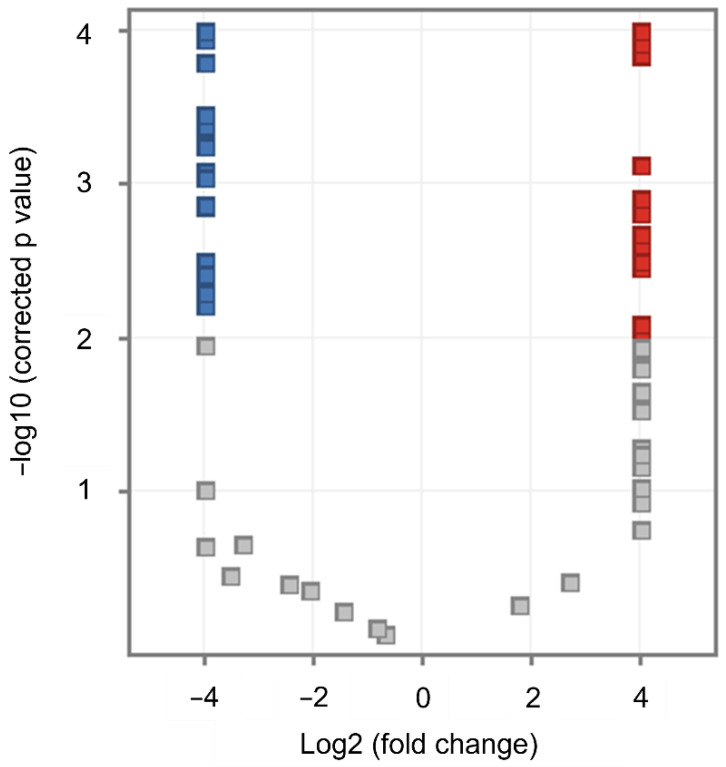
Volcano plot of MCF-7 vs. MDA-MB-231. 143 compounds are differentially found between both cell lines. The figure shows significant upregulated (red) and downregulated (blue) features (*p*-value ≤ 0.01); features that did not meet fold change or *p*-value are shown in gray.

**Table 1 ijms-23-12074-t001:** Identified compounds found with differential fold change by compound type and class of MCF-7 vs. MDA-MB-231.

Lipid Class	Lipid Type	Lipid	Dysregulation Tendency
Glycerophospholipids	Phosphatidylcholine	PC(18:1(11Z)/14:1(9Z))	Up
		PC(20:4(8Z,11Z,14Z,17Z)/22:5(7Z,10Z,13Z,16Z,19Z))	Down
	Lysophosphatidylcholine	LysoPC(17:0)	Up
		LysoPC(20:4(8Z,11Z,14Z,17Z))	Down
	Phosphatidylethanolamine	PE(20:2(11Z,14Z)/14:0)	Up
		PE(16:0/14:0)	Up
		PE(18:3(6Z,9Z,12Z)/20:3(8Z,11Z,14Z))	Up
		PE(18:2(9Z,12Z)/14:0)	Up
		PE(18:3(9Z,12Z,15Z)/18:3(6Z,9Z,12Z))	Up
		PE(14:1(9Z)/20:2(11Z,14Z))	Up
		PE(18:2(9Z,12Z)/20:3(5Z,8Z,11Z))	Up
		PE(20:3(8Z,11Z,14Z)/P-18:1(11Z))	Down
		PE(18:4(6Z,9Z,12Z,15Z)/P-18:0)	Down
		PE(20:5(5Z,8Z,11Z,14Z,17Z)/P-18:0)	Down
		PE(P-18:1(9Z)/22:5(7Z,10Z,13Z,16Z,19Z))	Down
		PE(P-18:0/16:0)	Down
		PE(P-18:1(9Z)/20:5(5Z,8Z,11Z,14Z,17Z))	Down
		PE(P-18:1(9Z)/22:6(4Z,7Z,10Z,13Z,16Z,19Z))	Down
		PE(P-18:1(9Z)/18:0)	Down
		PE(P-18:0/14:0)	Down
	Lysophosphatidylethanolamine	LysoPE(18:0/0:0)	Up
		LysoPE(18:1(9Z)/0:0)	Up
		LysoPE(22:6(4Z,7Z,10Z,13Z,16Z,19Z)/0:0)	Down
Sphingolipids	Ceramides	Cer(d18:0/24:0)	Up
		Cer(d18:0/26:1(17Z))	Up
			Up
		Lactosylceramide (d18:1/24:0)	Down
		Trihexosylceramide (d18:1/16:0)	Down
		Trihexosylceramide (d18:1/24:0)	Down
	Sphingomyelin	SM(d18:0/18:1(9Z))	Down
Glycerolipids	Monoradyglycerol	MG(18:0e/0:0/0:0)	Down
	Triglycerides	TG(16:0/18:3(9Z,12Z,15Z)/22:0)	Up
		TG(14:1(9Z)/15:0/22:2(13Z,16Z))	Down
Otros		Heptadecanoyl carnitine	Up
		beta-Sitosterol palmitate	Up
		Sphingosine	Up
		Isopeonidin 3-rutinoside	Up
		1,2,4-Nonadecanetriol	Down
		Propylene glycol stearate	Down
		AS 1-5	Down
		CE(22:6(4Z,7Z,10Z,13Z,16Z,19Z))	Down
		20,24-Epoxy-25,26-dihydroxydammaran-3-one	Down
		Bullatin	Down

## Data Availability

The data presented in this study are openly available in FigShare associated to https://doi.org/10.6084/m9.figshare.21153208.

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
