# Peer review of "LC–MS Based Lipidomics Depict Phosphatidylethanolamine as Biomarkers of TNBC MDA-MB-231 over nTNBC MCF-7 Cells"

_ijms, 2022, doi:10.3390/ijms232012074_

Round 1

Reviewer 1 Report (Previous Reviewer 4)

The authors have revised the manuscript relative to may earlier concerns. In general, the manuscript has been significantly improved. I have no further comments.

Author Response

Reviewer 2 Report (Previous Reviewer 3)

Although the manuscript improved a lot, the authors have not attached the list of the most prominent lipids up and down regulated or I am not able to see it.

Author Response

Reviewer 3 Report (New Reviewer)

In the current research, Ruben et al. focused on lipidomics analysis of MCF-7 and MDA-MB-231 breast cancer cell lines. The authors demonstrated that lipid synthesis influences the aggressiveness and treatment challenges of TNBC. In terms of articulating the work's reasoning, the research is quite weak and requires several revisions. The purpose of the investigation is not entirely evident. Nonetheless, please see my remarks below.

This raises the issue of why just these two cell lines were selected in the context of breast cancer. Regarding having the triple negative and not having the triple negative, there are many more alternatives. Therefore, it is vital to describe the difference in the other cell line type if a different cell line was employed in the same cellular setting.

In the last paragraph of the introduction, it would benefit the study if the authors explained the rationale for the work.

 In the method section, describe the sonication parameters, including the employed amplitude and ON and OFF pulses.

 In the results and discussion section, there is no mention of the experiment's reason. Please begin the results section with an explanation of the biological topic addressed by the experiment, why the authors chose the cellular model system, etc. Try to disregard the excessive wage sentences.

 Please modify Line 187, The identities, to a technical phrase, since it is not a biological term utilized in metabolomics research.

 The writing in the results and discussion part is quite bad. Please make the necessary adjustments to enhance readability.

Round 2

Reviewer 3 Report (New Reviewer)

No more comments

This manuscript is a resubmission of an earlier submission. The following is a list of the peer review reports and author responses from that submission.

Round 1

Reviewer 1 Report

Congratulations. Please show a validation of the data with other methods besides LCMS, which was the discovery step.

Reviewer 2 Report

This paper is too preliminary for publication. 

Reviewer 3 Report

In the manuscript entitled "LC-MS based lipidomics depict phosphatidylethanolamine as biomarkers of triple negative breast cancer cells" the authors try to find significant differences in common breast cancers (BC) and so called triple negative breast cancers (TNBC). They used two well-known cell lines MCF7 and MDA-MB-231 for their experiments modelling the common BC and TNBC, respectively.

The reviewer comments are the next.

- Basically the work is contributing to characterization of  the two well-known cell lines, therefore the title is exaggerated. I would suggest to change it for LC-MS based lipidomics depict phosphatidylethanolamine as biomarkers of MDA-MB-231 over MCF7, or something similar. The authors cited Purwaha et al., 2018 correctly, who describe a lipidomic analysis of 70 tumours. That work is much more comprehensive than this manuscript.

- My guess the work was done on MDA-MB-231 and not on MDA-MD-231 cells (lines of 20, 53, 193). If yes, please introduce and characterize as a new cell line.

- The authors did not present the full lipidomic data. At least they should have classified as Purwaha et al., 2018 did. The DATA NOT SHOWN (line 144) is not acceptable at this case, because this would be the meaning of the article.

- They should employ the negative ionization mode MS analysis along the positive one. Several lipids are detectable only in negative mode.

- I cannot see the importance of the Fig 1, and Fig2. Fig.4. They did not show more, than the Fig3. Although it is in contradiction with the sentence: We found 944 identities for MDA-MB-231 and 1007 cells for MCF-7. (line 142). According to Fig.3 there are only 505 entities for MDA-MB-231 and 534 for MCF7.

-The discussion is exaggerated similarly to the title. The results do not supports it. The sphingomyelins should be mentioned, at least, because there are data its role in TNBC.

Reviewer 4 Report

The manuscript reports a potentially important and interesting data. However, the paper has notable weaknesses. It seems to me that much more work is needed to improve the scientific values of the presented results. 

Specific comments/suggestion:     1.Title and conclusion of the manuscript suggest that phosphatidylethaloamine could be potential biomarker of triple-negative breast cancer cells (TNBC). However, insufficient  data are presented to confirm this suggestion. Conclusion is based only on differences in amount of lipids in MCF-7 (non TNBC) and MDA-MD231 (TNBC) cells. It is well known the abundance of phosphatidylethaloamine  varies in the membranes of different tissue and normal cells (non cancer cells) in mammals (Int Rev Cell Mol Biol 2016, 321, 29-88). Thus, there is no direct evidence that phosphatidylethaloamine could be associated only with TNBC.   2. Since  the research is incomplete at this version, it seems to me that need to explore why phosphatidylethaloamine could be associated with “ aggressivity  and difficulties of the treatment of TNBC”. Given that PE (phosphatidylethaloamine) binding protein 4 (PEBP4) is a key factor in the malignant proliferation  and metastasis of tumor cells (Mi Y J Thoracic Disease 2015; BBA 2016 1863, 1682-1689), the authors should discuss this problem.